# Improving wellness: Defeating Impostor syndrome in medical education using an interactive reflective workshop

Dotun Ogunyemi[1]*, Tommy Lee[1], Melissa Ma[2], Ashley Osuma[3], Mason Eghbali[2], Natalie Bouri[2]

1 Arrowhead Regional Medical Center, Colton, California, United States of America, 2 California University of Science and Medicine, Colton, California, United States of America, 3 University of Illinois College of Medicine, Peoria, IL, United States of America

* dogunye@outlook.com

## Abstract

### Background

Impostor syndrome is characterized by fraudulent self-doubt and correlates with burnout, and adverse mental health.

### Objective

The objective was to investigate correlates of Impostor syndrome in a medical education cohort and determine if an interactive workshop can improve knowledge and perception of Impostor syndrome.

### Methods

From June 2019 to February 2021 interactive educational workshops were conducted for medical education cohorts. Participants completed baseline knowledge and Impostor syndrome self-identification surveys, participated in interactive presentations and discussions, followed by post-intervention surveys.

### Results

There were 198 participants including 19% residents, 10% medical students, 30% faculty and 41% Graduate Medical Education (GME) administrators. Overall, 57% were positive for Impostor syndrome. Participants classified as the following Impostor syndrome competence subtypes: Expert = 42%; Soloist = 34%; Super-person = 31%; Perfectionist = 25%; and Natural Genius = 21%. Self-identified contributors of IS included: parent expectations = 72%, female gender = 58%, and academic rat race = 37%. GME administrators compared to physicians/medical students had significantly higher number of self-identified contributors to Impostor syndrome. Knowledge survey scores increased from 4.94 (SD = 2.8) to 5.78 (2.48) post intervention (p = 0.045). Participants with Impostor syndrome competence subtypes had increased perceptions of Impostor syndrome as a cause of stress, failure to reach full potential, and negative relationships/teamwork (p = 0.032 -<0.001).

**Data Availability Statement:** All relevant data are within the manuscript and its Supporting Information files.

**Funding:** There authors received no specific funding for this work.

**Competing interests:** The authors have declared that no competing interests exist.

## Conclusion

Impostor syndrome was common in this medical education cohort, and those with Impostor syndrome significantly attributed negative personal and professional outcomes to Impostor syndrome. An interactive workshop on Impostor syndrome can be used to increase perceptions and knowledge regarding Impostor syndrome. The materials can be adapted for relevance to various audiences.

## Introduction

Impostor syndrome (IS) is a psychological term characterized by chronic feelings of self-doubt and internalized fear of being exposed as an intellectual fraud. Sufferers are unable to internalize and own their successes, accomplishments, competence, or skills [1]. People with IS struggle with accurately ascribing their performance to their actual competence and tend to attribute successes to external factors such as luck or receiving help from others and attribute setbacks as evidence of their professional inadequacy [2,3]. Impostor syndrome was first identified by Dr. Pauline Clance in 1978 who recognized a state of "intellectual phoniness" that she had identified in a sample of high-achieving women [4].

Valerie Young ED is a leading expert on IS who created the Rethinking Impostor Syndrome™ which has delivered educational solutions including presentations, workshops, and coaching protocols regarding IS to over half a million people around the world since 1982. She developed the Young Impostor Scale (YIS) which is used to dichotomously assess for the presence or absence of IS [5]. YIS is widely used in the lay community and available on the internet. It was recently validated in a study of 138 medical students which revealed that almost a quarter of male medical students and nearly half of female students experienced IS and IS was found to be significantly associated with burnout indices [6]. In her book, "The Secret Thoughts of Successful Women: Why Capable People Suffer from the Impostor Syndrome and How to Thrive in Spite of It"; Dr. Young based on decades of research studying fraudulent feelings among high achievers uncovered five "competence subtypes"—or internal rules that people who struggle with confidence attempt to follow that may be holding them back from achieving their full potential. The competence subtypes are the perfectionist, the super-person, the natural genius, the soloist, and the expert [7].

While IS was initially identified among high-achieving professional women, contemporary research has documented these feelings of inadequacy among men and women, in many professional settings, and among multiple ethnic and racial groups [4]. A review on IS from 2019 found that many Internet users interacted with articles published online on IS over platforms such as LinkedIn, Forbes, and Medium.com [3]. This increased discussion of Impostor Syndrome in social media suggests that IS may currently be impacting many people.

Impostor syndrome has also been found to have implications on mental health. A review from 2020 revealed statistically significant associations between IS and components of mental health including self-esteem, psychological distress, burnout, anxiety, low self-esteem, and depression.(2) Furthermore IS has been shown to have implications for employment; career retention, job performance, and ultimately career advancement with working professionals questioning their legitimacy and qualifications [8].

Impostor syndrome has been described in medical students, internal medicine residents, family medicine residents, dental, pharmacy, nursing students, and clinical nurse specialists [9]. However, there is limited literature on contributors to Impostor syndrome and the role of educational interventions in a medical education population [10].

Therefore, the aims of our study were to 1) to investigate the prevalence of Impostor syndrome and competence sub-types in a medical education cohort, 2) to determine social, demographic, and professional risk factors that contribute to IS, and 3) to determine if a reflective and interactive educational workshop can improve awareness, perceptions, and knowledge regarding Impostor syndrome.

## Methods

### Study design

This was a retrospective cross-sectional study of interactive educational workshops conducted from June 2019 to February 2021.

### Curriculum development

We used Kern's six-step approach for curriculum development [11]. Step 1 is Problem Identification & General Needs Assessment: our literature review provided the rationale for the curriculum and enabled us to focus on meaningful goals and objectives. The conceptual framework identified and utilized was "situated learning-guided participation" in which didactic and interactive activities facilitate independent learning [12]. Step 2 is the Targeted needs assessment: we identified the specific needs and preferences of our targeted learners which included medical students, residents, faculty and staff and our specific learning environment via group discussions, institutional surveys and informal interviews with several residents and medical students. Step 3 is Goals and Objectives: we developed specific & measurable objectives regarding Impostor Syndrome. We hypothesized that demographic, social, and professional factors may be correlated with Impostor syndrome and that a workshop can improve short-term knowledge and perceptions. Step 4 are the Educational Strategies: to accomplish our educational objectives we identified the appropriate survey tools and developed the interactive curriculum. Step 5 is the Implementation: making the curriculum a reality and converting a good plan into an accomplishment. We identified resources, obtained some institutional support, and developed procedural processes to support the curriculum. Step 6 is Evaluation and feedback, which was accomplished by post intervention knowledge, perception, and behavior-based surveys. The evaluation was done using Kirkpatrick's framework at Kirkpatrick's Level 1: Reaction and Kirkpatrick's Level 2: Learning [13].

### Procedures

'The workshop was developed by the academic faculty and was facilitated by four faculty members, two residents and two medical students. Participants included medical students, resident physicians, faculty, and graduate medical education (GME) coordinators referred to as academic administrators. The interactive and reflective educational workshop was approximately 60–90 minutes in duration. The workshop included 1) pre-survey; 2) animated video presentation; 3) PowerPoint interactive presentation; 4) small group discussion on mitigating Impostor in the learning environment; 5) whole group discussion "You can still have an impostor moment, but not an impostor life". The small and large group exercises and discussions enabled participants to identify corrective strategies and 6) a post- intervention survey.

### Instruments

In the pre-intervention phase of the workshop, participants first completed a baseline knowledge test consisting of twelve items on IS. Participants then completed the eight-item

assessment Young Impostor Syndrome (YIS) instrument. Five out of eight positive responses are considered a positive result for IS. Participants further completed a survey to identify their competence subtypes (super-person, soloist, natural genius, expert, perfectionist), which is how individuals with IS may perceive competence. Experts identify competency as ability to master tasks quickly. Soloists identify competency as able to master tasks without seeking help or guidance from others. Geniuses question their competency if they make any small mistake while completing tasks. Super-persons question their competency if they are unable to succeed in every single aspect of their life. Perfectionists will only attempt tasks that they are certain they can master and can execute flawlessly. Each competence subtype was confirmed if the participant had a mean score of four or five from a range of 1–5 [5,14]. In the post-intervention portion, participants completed perception surveys and post-intervention knowledge assessment of IS (S1–S3 Appendix).

## IRB

Informed consent was waived by IRB since this was a retrospective study and all data accessed was fully anonymized. Data was accessible only to the research team. Informed consent was not obtained at the time of the interactive educational workshops because the workshops were conducted in established or commonly accepted educational settings with information obtained in such a manner that the identity of the human subjects could not be readily ascertained, directly or through any identifiers linked to the subjects. Anonymous, private surveys were used to minimize the risks of participation. The choice to participate or to complete any of the survey tools was voluntary and had no impact on the learners' standing in their educational program. Institutional Review Board (IRB) exempt approval was obtained.

## Participants

This study utilized a convenient sample of a medical education cohort who attended various workshops that was presented in multiple voluntary sessions to medical students, residents, faculty and staff in internal medicine, family medicine, psychiatry, surgery, obstetrics and gynecology departments at California University of Science and Medicine and Arrowhead Regional Medical Center, Colton California. The workshop was also presented at the 2021 Accreditation Council for Graduate Medical Education (ACGME) Annual Conference. Graduate Medical Education (GME) residency program coordinators also referred to as administrators participated in the workshops.

## Analysis

Statistical analysis was performed using SPSS 21.0 (IBM Corp, Armonk, NY). Student's t tests, chi square test and ANOVA were performed as indicated including the calculation of 95% confidence interval and odds ratio. A two-sided P value <0.05 was accepted as significant. The frequencies of IS and IS competency subtypes were calculated. Analysis was completed to identify significant associations between IS and demographic factors (age, gender, ethnicity, employment position), IS competency subtypes, and contributory factors (parental expectations, female gender, academic rat race, first generation American, first-generation college, transitions, underrepresented minorities, mental health) and perception of the adverse impacts of IS. We also conducted analysis to identify significant differences between academic administrators and physicians/medical students. Underrepresented in medicine racial groups (UiM) included Hispanics, African American, Native American.

## Results

Of 198 participants, 178 respondents (90% response rate) completed the surveys, including 138 females and 40 males. Only 16.4% of respondents had received prior educational training on IS. There were 18 medical students (10.1%), 33 resident physicians (18.5%), 22 physician faculty (12.4%), 74 academic administrators (41.6%), and 31 (17.4%) program directors. Sixty-six percent of physicians were primary care providers. In this medical education group, 102 participants (57%) were found to be positive for Impostor syndrome as measured by the Young Impostor Syndrome Scale. Of the Impostor Syndrome competence subtypes, the most common subtype was expert in 42%; 33.7% identified as Soloists, 30.7% as super-person, 25.5% as perfectionists, and 20.6% identified as Geniuses. Self-identified contributors of IS included: parent expectations = 72%, female gender = 58%, academic rat race = 37% and first generation to go to college = 35% (Table 1).

Participants who scored positive for IS also scored significantly higher on all perceptions regarding the adverse impact of IS on their personal and professional relationships but specifically scored significantly higher on recognizing IS as a source of stress, and a limitation on achieving their full potential. Of the five IS competence subtypes only three (expert, super person, and perfectionist) significantly correlated with scoring positive for IS in this cohort. Younger age was only significantly associated with IS on one-sided but not on the two-sided chi square test. Even though the incidence of IS was higher in underrepresented in medicine racial groups, the difference was not statistically significant (Table 2).

Baseline scores on the IS Knowledge Survey for all participants was 4.94 (SEM = 0.31) pre-intervention, which increased to 5.65(0.18) post-intervention (p = 0.045).

We compared GME administrators to physicians/medical students. This revealed that physicians/medical students had significantly higher scores on both the pre and post knowledge workshop scores than academic administrators. Furthermore, GME administrators had significantly higher mean scores of total numbers of self-identified contributory factors to IS and total perception scores for the adverse impacts of IS. Specifically, for contributory risk factors, GME administrators score significantly higher on career changes and physicians/medical students scored significantly higher on transitions. Physicians/medical students were significantly more likely to have had previous training in IS. There were no differences between the physicians/medical students and GME administrators regarding prevalence of IS (54% versus 63%); but for the five competence subtypes, GME administrators had significantly higher scores for soloist, natural genius and perfectionist, whilst physician/medical students had significantly higher scores for expert and super-person competence subtypes (Table 3).

We also performed analyses to determine significant associations of the five competence subtypes. Perfectionist, natural genius, and expert competence subtypes each were significantly identified as a source of stress, a limitation on achieving full potential and having negative impact on family relations. Natural genius and soloist competence subtypes were associated with a negative effect on team function. Other significant associations with contributory factors were perfectionist competence subtype with female gender and parents' expectations; natural genius competence subtype with mental health disorders; Expert with GME administrators; whilst soloists were less likely to be associated with academic rat race and super-person less likely to be present in the UiM group (Table 4).

## Discussion

In this cross-sectional study, we developed a curriculum for an Impostor Syndrome educational interactive workshop using the Kern's six-step approach. We used the framework grounded in situated learning and guided participation to develop interactive and engaging

**Table 1. Frequency of Impostor syndrome and competency subtypes, demographics, other characteristics, and contributory factors as identified by participants.**

| Characteristics (n = 178) | Number | Percentage |
|---|---|---|
| Academic Characteristics | | |
| GME administrators! | 74 | 41.6 |
| Program Directors/administrative leaders | 31 | 17.4 |
| Residents | 33 | 18.5 |
| Faculty | 22 | 12.4 |
| Medical students | 18 | 10.1 |
| Primary Care practice | 70 | 66 |
| Non-Primary care practice | 36 | 34 |
| Previous Training (n = 128) | 21 | 16.4 |
| Population Characteristics | | |
| Young (40 years or younger) (n = 128) | 60 | 46.9 |
| Born in United States (n = 127) | 111 | 87.4 |
| Female gender (n = 178) | 138 | 77.5 |
| Non-Hispanic White (n = 160) | 96 | 60 |
| Asian (n = 160) | 39 | 24.4 |
| Latinx | 15 | 9.4 |
| UiM (n = 160) | 27 | 16.9 |
| Diagnosis and Competencies (n = 179) | | |
| Impostor syndrome | 102 | 57 |
| Expert | 81 | 42.2 |
| Soloist | 64 | 33.7 |
| Natural genius | 39 | 20.6 |
| Super person | 58 | 30.7 |
| Perfectionist | 48 | 25.5 |
| Risk (Contributory) factors to IS | | |
| Parents expectations | 131 | 72.4 |
| Female gender | 103 | 57.5 |
| Rat race | 66 | 36.5 |
| 1st generation in college | 64 | 35.4 |
| Change of careers | 63 | 34.8 |
| Mental health issues | 42 | 33.1 |
| Transitions | 48 | 26.5 |
| UiM | 41 | 22.7 |
| Unsupportive work culture | 37 | 20.4 |
| LGBTQ | 18 | 14.2 |
| 1st generation American | 15 | 11.8 |
| English as a second language | 13 | 10.2 |
| International transition | 18 | 9.9 |
| Foreign accent | 11 | 8.7 |
| Disability | 6 | 4.7 |
| Other | 32 | 25.2 |

UiM = Underrepresented in medicine racial groups: Hispanics, African American, Native American.

! GME Administrator, also known as GME program coordinator.

Primary Care practice = Internal Medicine, Family Medicine, Pediatrics, Obstetrics & Gynecology.

**Table 2. Associations of participants who were identified with impostor syndrome versus those who were not.**

| Variable | Impostor syndrome (n = 107) | No Impostor syndrome (n = 71) | P value (OR 95%CI) |
|---|---|---|---|
| Female | 80 (60.6%) | 52 (39.4%) | 0.068 |
| UiM* | 14(56%) | 11(44%) | 0.83 |
| Young age (≤ 40 years) | 38 (37.3%) | 18 (23.4%) | 0.034; 1.22, 1.01 to 1.48! |
| Expert | 52 (52.5%) | 21 (28.4%) | 0.002; 2.79, 1.47 to 5.30 |
| Super person | 41 (41.4%) | 13 (18.1%) | 0.001; 3.21, 1.56 to 6.60 |
| Perfectionist | 35 (35.4%) | 8 (11.3%) | <0.001; 4.031; 1.85 to 10.01 |
| Natural genius* | 25(25.3%) | 10(13.9%) | 0.085 |
| Soloist* | 37 (37.4%) | 21 (29.2%) | 0.327 |
| Limited my potential | 4.09 (0.81) | 3.42 (1.01) | < .0001; 0.38 to 0.95 |
| Source of stress | 4.12(0.75) | 3.48 (1.06) | < .0001; 0.35 to 0.92 |
| Total Perception mean score | 19.11 (3.27) | 17.12 (4.49) | 0.003; -3.28 to -0.70 |

(): standard deviation.

! This was only significant on one-sided t test but not on 2-sided t-test which was 0.052.

* UiM, natural genius and soloist were not statistically significant.

UiM = Underrepresented in medicine racial groups: Hispanics, African American, Native American.

activities including small and large group discussion sessions to promote interests in the concepts and principles regarding IS, support independent learning and encourage problem solving. However, we do not have any longitudinal data to confirm that the independent learning continued over time. We also evaluated our program only at Kirkpatrick's reaction and learning levels; evaluations at Kirkpatrick's behavior level (the degree to which participants apply what they learned during training when they are back on the job) may be more useful.

In this medical education cohort, more than half of the participants (57%) were positive for Impostor syndrome. These individuals compared to those negative for IS had higher perceptions of the adverse impact of IS on reaching their full potential and professional relationships.

**Table 3. Significant findings of comparisons between GME administrators (n = 71) versus physicians and medical students (n = 75).**

| Variable | Physicians/medical students | GME Administrator! | P value; (OR; 95%CI) |
|---|---|---|---|
| Young age (< 40 years) | 25 (24%) | 33 (44.6%) | 0.006; 1.86, 1.21 to 2.84 |
| Previous training | 12 (23.5%) | 7 (9.7%) | 0.045; OR, 2.86,1.04 to 7.87 |
| Risk factor: Transitions | 31 (31.3%) | 9(13.2%) | 0.009; 2.99, 1.31 to 6.79 |
| Risk factor: Changed careers | 22 (22.2%) | 33 (48.5%) | <0.001; 0.30; 0.16 to 0.59 |
| Pre workshop knowledge score | 5.67 (3.09) | 3.78 (1.73) | 0.008; 0.50 to 3.29 |
| Post workshop knowledge score | 6.65 (2.44) | 4.82 (2.16) | <0.001; 1.08 to 2.59 |
| Risk (contributory) factors | 3.48 (2.27) | 4.19 (2.16) | 04; -1.39 to -1.01 |
| Perceptions mean score | 10.77 (2.78) | 12.23 (1.73) | <0.001; -3.75 to -1.35 |
| Experts mean score | 9.52 (4.03) | 7.34 (2.14) | <0.001; 1.1 to 3.22 |
| Soloists mean score | 9.71 (2.58) | 10.54 (2.17) | 0.027; -1.55 to -0.09 |
| Natural genius mean score | 15.91 (3.67) | 17.06 (3.15) | 0.032; -2.20 to -0.10 |
| Super people mean score | 14.06 (4.08) | 12.67 (3.45) | 0.018; 0.24 to 2.56 |
| Perfectionists mean score | 12.67 (3.45) | 14.30 (2.66) | 0.001; -2.58 to -0.71 |

(): standard deviation.

! GME Administrator, also known as GME program coordinator.

**Table 4. Significant associations of the impostor syndrome competence subtypes.**

| Variable | Perfectionist | Not Perfectionist | P value (OR 95%CI) |
|---|---|---|---|
| Risk factor: Parents expectations | 40 (87%) | 88 (67.2%) | 0.012; 3.258, 1.28 to 8.28 |
| Risk factor: Female gender | 35 (76.1%) | 65 (50.4%) | 0.003; 3.13; 1.47 to 6.70 |
| Source of stress | 4.32 (0.52) | 3.77 (1.02) | 0.001; -0.86 to -0.23 |
| Limited my potential | 4.27 (0.086) | 3.73 (1.02) | 0.001; -0.87 to -0.22 |
| Negative effect on family relations | 4.02 (0.59) | 3.54 (1.18) | 0.01; -0.85 to -0.12 |
| Risk factors total score | 4.54(2.13) | 3.69 (2.85) | 0.029; -1.59 to -0.11 |
| Perception total score | 19.75 (2.32) | 18.04 (4.12) | 0.01; -3.02 to -0.40 |
| Variable | Natural genius | Not natural genius | P value (OR 95%CI) |
| Risk factor: Mental health issues | 16 (57.1%) | 26 (26.9%) | 0.006; 3.64, 1.52 to 9.71 |
| Limited my potential | 4.38 (0.70) | 3.74 (0.97) | <0.001; -0.93 to -0.35 |
| Source of stress | 4.41 (0.61) | 3.78 (0.98) | <0.001; -0.89 to -0.36 |
| Negative effect on team function | 4.09 (0.83) | 3.65 (1.00) | 0.011; -0.77 to -.0.10 |
| Negative effect on family relations | 4.06 (0.85) | 3.57 (1.10) | 0.017; -0.89 to -0.09 |
| Learned skills to deal with IS | 4.38 (0.65) | 4.11 (0.73) | 0.036; 053 to -0.02 |
| Perception total score | 20.26 (2.81) | 18.04 (3.94) | <0.001; -3.40 to -1.06 |
| Behavior changes total score | 8.41 (1.35) | 7.86 (1.54) | 0.044; -1.08 to -0.016 |
| Variable | Expert | Not Expert | P value (OR 95%CI) |
| Academic administrators | 34 (55.7%) | 36 (38.7%) | 0.047; 1.39; 1.001 to 1.92 |
| Source of stress | 4.15 (0.83) | 3.74 (0.98) | 0.003; -0.69 to -0.14 |
| Limited my potential | 4.17 (0.84) | 3.65 (0.98) | <0.001; -0.79 to– 0.25 |
| Negative effect on family relations | 3.88 (1.17) | 3.52 (1.05) | 0.32; -0.68 to -0.32 |
| Knowledge total pre score | 4.17 (2.71) | 5.61 (2.73) | 0.022; 0.22 to 2.67 |
| Perception total score | 19.42 (3.24) | 17.80 (4.08) | 0.006; -2.75 to -0.47 |
| Variable | Soloist | Not Soloist | P value (OR 95%CI) |
| Risk factor: Rat race | 14 (23.7%) | 51 (42.5%) | 0.02; 0.42, 0.21 to 0.85 |
| Negative effect on team function | 3.97 (0.90) | 3.63 (1.01) | 0.033; -0.65 to -0.027 |
| Negative effect on family relations | 3.95 (0.96) | 3.53 (1.11) | 0.014; -0.76 to 0.085 |
| Underrepresented minorities | 6 (10.2%) | 35 (29.2%) | 0.004; 0.28; 0.11 to 0.70 |
| Variable | Super person | Not Super person | P value (OR 95%CI) |
| Source of stress | 4.13 (0.83) | 3.81 (0.99) | 0.027; -0.61 to -0.04 |

(): Standard deviation.

Contributory (risk factors) identified by the participant as a cause of their Impostor syndrome analyzed in this table included: Parents expectations, Female gender, Rat race, 1st generation American, 1st generation in college, transitions, UiM, and mental health.

Limited my potential = Perception question: I now realize that I have Impostor syndrome that has impeded me from rising to my full potential.

Source of stress = Perception question: It is clear to me that my Impostor syndrome may be responsible for stress, low self-esteem, frustration, or other negative emotions in my life.

Negative effect on team function = Perception question: I can see how my Impostor syndrome may have negatively affected my team at work or school.

Negative effect on family relations = Perception question: I now realize how my Impostor syndrome may have affected my relationships and my family.

Learned skills to deal with IS = Behavior question: I feel that I have learned some skills from this workshop that can help me deal with my Impostor syndrome.

Baseline scores on the IS Knowledge Survey for all participants who participated in the workshop also increased significantly from pre-intervention mean of 4.94 to post-intervention mean of 5.78.

Existing scholarly work on Impostor syndrome demonstrates that IS has been documented in many populations, with varying prevalence. A systematic review from 2020 reported that the prevalence of IS ranged from 22.5% to 46.6% amongst medical residents, physicians, and

medical students [2]. Our findings of a prevalence of 57%, is slightly higher than previously published. This may be due to our small population size, different diagnostic criteria, or specific population characteristics. However, our findings may also reflect the increased relevance of IS which is in alignment with the noted considerable lay interest in IS based on social media postings. For example, Bravata et al. noted 133,425 engagements (e.g., "likes," re-postings, comments) on social media platforms such as Facebook and Twitter from March 28, 2018–March 18, 2019 [3]. The percentage of students meeting criteria for the impostor syndrome in our study was not significantly different between males and females. This is consistent with the literature; whereas the earlier literature on impostor syndrome focused on women, half of the included studies in a recent systemic review found no difference in the rates of men and women suffering from impostor syndrome [3]. In our study there were no significant differences between IS rates in UiM (56%) versus non-UIM participants (44%). Previous studies have demonstrated that IS was more common among underrepresented in medicine racial groups, with impostor feelings significantly negatively correlated with psychological wellbeing and positively correlated with depression and anxiety. [2,3,15].

Previous studies in medical education regarding IS have focused on students and healthcare providers, with none noted on GME administrators [2,3,6]. GME administrators work behind the scenes in residency programs providing crucial logistical, and administrative support to residents, program directors, faculty, and staff; and maintaining databases and processes to meet Accreditation Council of Graduate Medical Education (ACGME) accreditation. Although GME administrators play an integral role in residency training programs, there is a paucity of data regarding risk factors that predict their burnout and wellness [16,17]. In this study, the prevalence rate of IS of approximately 64% for GME administrators was higher than that for physicians/medical students at 54% but was not statistically significantly different. GME administrators identified significantly higher rates of contributory risk factors for IS including the negative impact of IS on personal and professional relations. This data adds to a recent study demonstrating that 74% of family medicine residency coordinators had moderate to high burnout rates with a significant negative correlation between overall job satisfaction and burnout scores (16). In our study, GME coordinators significantly identified career changes as a significant contributory risk factor to IS. Career changes suggests increased turnover rates which has been shown to be negatively correlated with job satisfaction, and burnout (16). This data demonstrates that GME administrators are at increased risk of IS similar to faculty, residents, and medical students. Therefore, GME coordinators/administrators should be actively and strategically included in institutional and GME wellness programs in order to increase work satisfaction and decrease burnout rates.

Competence subtypes described by Dr. Valerie Young which include expert, super person, perfectionist, soloist, and natural genius refer to the unconscious rule of how individuals define what it means to be competent. Dr. Young explained that all 5 competence subtypes hold an extreme view of competence, with no perception of a competence middle ground and if the person is not operating at the top of their game 24/7, they view themself as incompetent [7,14,18,19]. Though there is a significant amount of literature on IS, a literature review on PubMed using key words of "Impostor", "competence types", "Impostorism", "Impostor" did not reveal any published studies on the Impostor competence subtypes. However considerable lay interest was noted with sites devoted to descriptions and analyses of Impostor competence subtypes [14,19,20]. In this academic medical cohort, expert, and soloist competence subtypes were the most common. Overall, Impostor competence subtypes significantly correlated with adverse perceptions such as stress, limitation of full potential, negative family, and team relations. Contributory risk factors were also significantly associated with competence subtypes. Parent expectations and female gender were three times more likely to be associated with

those with perfectionist competence subtypes, while mental health issues were three times more likely in those scoring positive for the natural genius subtype. GME administrators had significantly higher scores for soloist, natural genius, and perfectionist, whilst physician/medical students had significantly higher scores for expert and super-person competence subtypes. However, there were some incongruencies with YIS tool since only three of the five IS competence subtypes (expert, super person, and perfectionist) significantly correlated with scoring positive for IS in this cohort. Therefore, we consider our findings on competence subtypes as preliminary and pilot that would require confirmation from future studies. Furthermore, even though the competence subtypes instrument tool is in extensive use in the lay community nonetheless we could not find any published validation data. Our findings even though introductory and requiring confirmation have practical implications since there are different strategies, counseling, and practical interventions to combat and mitigate each competent subtype [5,7]. For example, the super person can be taught to reframe failure as a learning opportunity, while the perfectionist can be coached to "push yourself to act before you are ready' and the expert counseled to avoid unequal comparisons with people who have more experience in a role.

Publication on effective tools to intervene and confront IS are limited. Rivera et al published on a 75-minute workshop that involved individual reflection, small-group case discussion, and large-group instruction and concluded that the workshop was an effective means to discuss strategies on how to address IS at the individual, peer, and institutional levels(1). Other published educational interventions include a facilitator-guided 30- to 45-minute intervention in internal medicine residents, a three-hour educational session with new internal medical interns, an educational workshop with clinical nurse specialist students and an online module consisting of a 14-minute educational video to dental students [21–24]. In comparison to the previous studies, our study had a more robust study population of 178 compared to 98 participants by Rivera and 21 by participants by Baumann [1,24]. Our study builds on the previous studies by not only showing an increase in knowledge of IS but also enabled participants to reflect on risk factors that predisposed to them to IS and the impact of IS on personal, social, and professional outcomes.

Our study also used the Young Impostor Scale (YIS) to identify participants with IS. A review of the academic literature only revealed the one publication by Villwock et al using YIS with medical students. The majority of published studies used the Clance Impostor Phenomenon Scale (CIPS), which is a 20-item five-point Likert scale tool which is scored in 4 categories of low (≤ 40), moderate (41–60), high (61–80), and intense (>80) IS. In comparison, the Young Impostor Scale is an eight-item scale, in which a diagnosis of Impostor syndrome is made with a positive response to five of the eight questions. Consequently, the YIS is simpler and less time consuming to administer and to score. However, it has been noted that there is a lack of normative data for the Young Impostor Scale and that a normal or acceptable level of IS has not been delineated [6]. Our study further adds to the literature by using YIS not only with medical students but also with residents, faculty, and academic coordinators.

Limitations of this study include that it is a self-administered, survey based, convenience sample of a medical education cohort. Furthermore, there are no robust validated or normative data on the survey tools utilized in this study. However, generalizability was increased because this study included participants from more than one institution as compared to most previous studies on IS. The self-reporting and self-selection of participants may have introduced a tendency to social desirability response bias. There is also a possible participation bias, or selection bias as participants were self-selected and not all completed the survey, however the response rate of 90% is very robust. It is possible that our small sample size may have resulted in a type 2 error which may explain the non-significant findings between IS and

underrepresented in medicine racial groups, gender, soloist, or natural genius competence subtypes. Another limitation is the cross-sectional design of the study which does not allow the establishment of causal relationships. Accordingly, the findings may not generalize to other times, schools, educational programs, or professional groups. It is also possible that our discussions and approach may reveal a more positivist lens bias to the constructs and does not adequately describe the uncertainty in measurement and in the construct.

## Conclusions

In conclusion, our study demonstrated that Impostor syndrome was prevalent within this medical education cohort, and that a reflective interactive workshop was useful in increasing knowledge and awareness regarding IS. Our study further adds to the literature by identifying significant differences between academic administrators and physicians/medical students; analyzing self-identified contributory factors and providing data on the IS competence subtypes. Future studies on Young Impostor Scale and the IS competence subtypes are required to better clarify the validity and utility of these instruments. This interactive and reflective workshop has the potential to be applied to other medical education settings. This information may enable future targeted IS interventions for specific groups within medical education community and overall lead to development of effective tools to mitigate Impostor syndrome impact and sequalae amongst sufferers especially in the clinical learning environment.

## Supporting information

**S1 Checklist.**
(DOCX)

**S1 Table. Session time-table and presentation.** Outline of reflective workshop schedule and PowerPoint Presentation.
(DOCX)

**S2 Table. Impostor syndrome competence subtypes.** Description of the 5 competence subtypes within Impostor syndrome.
(DOCX)

**S1 Appendix. Pre-test survey.** Survey materials completed by participants prior to reflective workshop.
(DOCX)

**S2 Appendix. Group activity.** Materials for Group activities utilized during reflective workshop.
(DOCX)

**S3 Appendix. Post-test survey.** Survey materials completed by participants after reflective workshop.
(DOCX)

## Author Contributions

**Conceptualization:** Dotun Ogunyemi.

**Data curation:** Melissa Ma, Ashley Osuma, Mason Eghbali.

**Formal analysis:** Dotun Ogunyemi.

**Investigation:** Melissa Ma, Ashley Osuma, Mason Eghbali.

**Methodology:** Tommy Lee.

**Supervision:** Dotun Ogunyemi.

**Validation:** Tommy Lee, Natalie Bouri.

**Writing – original draft:** Dotun Ogunyemi, Natalie Bouri.

**Writing – review & editing:** Natalie Bouri.

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
