## [Decision Letter · Decision Letter 0]

16 Jun 2022

PONE-D-22-00137Improving wellness: defeating imposter syndrome in medical education using an interactive reflective workshopPLOS ONE

Dear Dr. Ogunyemi,

Thank you for submitting your manuscript to PLOS ONE. After careful consideration, we feel that it has merit but does not fully meet PLOS ONE’s publication criteria as it currently stands. Therefore, we invite you to submit a revised version of the manuscript that addresses the points raised during the review process. The overarching theme is one of the manuscript needing to evolve and mature to more fully describe the methods and then discuss the desribed results in the context of the named theoretical framework to draw less causal conclusions. The work will likely be accepted on addressing these issues.

We look forward to receiving your revised manuscript.

Kind regards,

Dylan A Mordaunt, MD, MPH, FRACP

Academic Editor

PLOS ONE

**Journal requirements:**

3. PLOS requires an ORCID iD for the corresponding author in Editorial Manager on papers submitted after December 6th, 2016. Please ensure that you have an ORCID iD and that it is validated in Editorial Manager. To do this, go to ‘Update my Information’ (in the upper left-hand corner of the main menu), and click on the Fetch/Validate link next to the ORCID field. This will take you to the ORCID site and allow you to create a new iD or authenticate a pre-existing iD in Editorial Manager. Please see the following video for instructions on linking an ORCID iD to your Editorial Manager account: https://www.youtube.com/watch?v=_xcclfuvtxQ.

**Additional Editor Comments:**

Thank you for your submission.

- PLoS One is an interesting place to submit this piece to, entirely appropriate but studies on medical students are often seen in journale like BMJ Postgraduate Medicine. I think there's both a general and a specific audience, the latter including medical practitioners and clinical educators.

- In general, the manuscript could do with tightening up on referencing- I think definitions should be more clearly referenced, for instance.

- Kern’s six-step approach could be described initially, then clarified what the modifications, if any, were in this application of the method.

- The methods section could be split further into approach, data collection etc. Looking at checklists for publication of DELPHI studies might be a way to ensure all the areas are captured.

- The Young Imposter Instrument doesn't appear to be clearly referenced at introduction, and it should be made clear whether the tool is validated etc. The explaination is abrupt and appears incomplete- there's no word limit in PLoS One, so try elaborate concisely to make it clearer what the instrument measures.

- The way in which the discussion section is written could also be tightened up. I think the framing pushes into causality and takes a positivist approach to the constructs. It's not clear how the too brief theoretical framework described in the methods was applied in the interpretation of the results and I guess that's where I can see a bit of dissonance between stated epistemology/methodology and what appears to be a more positivist lens? I can't help but feel like a greater description of what the results mean is necessary before jumping into the assertion of what it demonstrated and the opening line of the discussion seems displaced. I acknowledge that this is a stylistic issue but we want your manuscript to be the highest quality and have the highest impact it can, and overall I feel the discussion can be improved, made more robust follow a more structured approach and argue for the conclusion you land on, piecing the collected evidence together with theoretical framework and available wider literature in discussion.

- Lastly, I feel that there is a bit of a tendency to lean towards discreet groups and concepts and that the uncertainty in measurement and in the construct, are not well reflected in the writing.

With specific regards to the criteria for publication:

1. The study appears to present the results of original research.

2. Results reported do no appear to have been published elsewhere.

3. Experiments, statistics, and other analyses are performed to a reasonable standard though I feel the description is lacking in completeness. A structured reporting tool would help with this.

4. Conclusions are not really presented in the best fashion, they appear abruptly at the onset of the discussion and so don't develop a logical argument leading into the conclusion; conclusions are presented causally, which I appreciate is both a stylistic and epistemologic issue, but worth considering as you approach your revision.

5. The article is presented in an intelligible fashion and is written in standard English. It is at times a little clumsy and this could be improved by ensuring continuity of thought and logic through the writing.

6. An IRB statement is present.

7. The article adheres to appropriate reporting guidelines and community standards for data availability, insofaras there aren't specific reporting guidelines that I'm aware of. However, as mentioned above I think a structured reporting tool could help flesh out the methods a bit better in particular. The study captures qualitative ideas but uses survey methods. The checklists I suggest looking at are CROSS (https://www.equator-network.org/reporting-guidelines/a-consensus-based-checklist-for-reporting-of-survey-studies-cross/), SRQR (https://www.equator-network.org/reporting-guidelines/srqr/) and COREQ (https://www.equator-network.org/reporting-guidelines/coreq/). I would specifically suggest leveraging CROSS, and including the completed form as a supplementary item.

I look forward to your resubmission. It's great to read a study that delves into some of the complexity of medical training and an important psychological element of it. I could see the study being repeated in the same cohort later in their career, or in a post-graduate cohort to try track how this obervations change with changes in seniority- interns, residents, fellows and consultants/attending.

Reviewers' comments:

Reviewer's Responses to Questions

**Comments to the Author**

1. Is the manuscript technically sound, and do the data support the conclusions?

Reviewer #1: Yes

2. Has the statistical analysis been performed appropriately and rigorously? 

Reviewer #1: No

3. Have the authors made all data underlying the findings in their manuscript fully available?

Reviewer #1: Yes

4. Is the manuscript presented in an intelligible fashion and written in standard English?

Reviewer #1: No

5. Review Comments to the Author

Reviewer #1: First, the writing needs to be improved, see sentence below and "affected" should be used instead of "impacted." SEM is not an appropriate statistical tool for a single study; SD needs to be used. I note an extreme sex/gender ratio bias in those completing the survey; is this reflective of the employee base of the institutions studied? Looking at this sentence:

"Furthermore, there were racial differences with administrators more likely to be underrepresented minority groups which may be reflection of systemic and institutional inequities and access to higher education."

I realize that different institutions have different employee cohorts, but most medical schools that I am aware of have an administration composed of MDs, PhDs, Eds, etc., these are the highest paid and highest ranking members of the institution, and make the decisions in running the institution. It is unclear how being a high status, highly paid administrator is reflective of inequities, unless that is meant to describe those employees who are not administrators. The sentence is also poorly written - "reflection" should be "reflective."

For the reader not well acquainted with these terms, can it be explained why positive terms like "genius" etc. (that would seem to be associated with high self-esteem) are associated with IS? Finally, a limitation of these studies in general is a lack of correlation with subjective self-evaluation and actual performance. IS implies that the person falsely believes that they are not competent. Whether or not this is true or false is not determined.

6. PLOS authors have the option to publish the peer review history of their article (what does this mean?). If published, this will include your full peer review and any attached files.

Reviewer #1: No

---

## [Author Response · Author response to Decision Letter 0]

19 Jul 2022

To:

Dylan A Mordaunt, MD, MPH, FRACP

Academic Editor

PLOS ONE

Re: PONE-D-22-00137. Improving wellness: defeating imposter syndrome in medical education using an interactive reflective workshop.

Thank you for allowing us to revise our manuscript. We have diligently reflected and carefully responded to each of the reviewers’ comments. We hope our work and response will be found worthy of publication in PLOS. Please let us know if more is required of us. 

Our responses are as follows:

Specific comments:

Comment:

Response:

We have reviewed the PLOS ONE's style requirements and ensure that we followed the instructions.

Comment:

Response:

The following has been included in the Methods section:

Line 153-163:

Informed consent was waived by IRB since this was a retrospective study and all data accessed was fully anonymized. Data was accessible only to the research team. Informed consent was not obtained at the time of the interactive educational workshops because the workshops were conducted in established or commonly accepted educational settings with information obtained in such a manner that the identity of the human subjects could not be readily ascertained, directly or through any identifiers linked to the subjects. Anonymous, private surveys were used to minimize the risks of participation. The choice to participate or to complete any of the survey tools was voluntary and had no impact on the learners’ standing in their educational program. None of the investigators had any conflict of interest and there was no extra funding, with investigators donating their time and expertise. Institutional Review Board (IRB) exempt approval was obtained.

Comment:

3. PLOS requires an ORCID iD for the corresponding author in Editorial Manager on papers submitted after December 6th, 2016. Please ensure that you have an ORCID iD and that it is validated in Editorial Manager. To do this, go to ‘Update my Information’ (in the upper left-hand corner of the main menu), and click on the Fetch/Validate link next to the ORCID field. This will take you to the ORCID site and allow you to create a new iD or authenticate a pre-existing iD in Editorial Manager. Please see the following video for instructions on linking an ORCID iD to your Editorial Manager account: https://www.youtube.com/watch?v=_xcclfuvtxQ.

Response:

We have followed these instructions

Comment:

Please review your reference list to ensure that it is complete and correct. If you have cited papers that have been retracted, please include the rationale for doing so in the manuscript text or remove these references and replace them with relevant current references. Any changes to the reference list should be mentioned in the rebuttal letter that accompanies your revised manuscript. If you need to cite a retracted article, indicate the article’s retracted status in the References list and also include a citation and full reference for the retraction notice.

Response:

We have re-arranged the references to align with the revisions to the manuscript. We have updated the links to online references and included the date accessed. 

Comment:

Thank you for your submission.

- PLoS One is an interesting place to submit this piece to, entirely appropriate but studies on medical students are often seen in journal like BMJ Postgraduate Medicine. I think there's both a general and a specific audience, the latter including medical practitioners and clinical educators.

Response:

Thank you for the favorable comments and the opportunity to publish our work in PLOS. We agree that this study may reach a wider audience in PLOS.

Comment:

- In general, the manuscript could do with tightening up on referencing- I think definitions should be more clearly referenced, for instance.

- Kern’s six-step approach could be described initially, then clarified what the modifications, if any, were in this application of the method.

Response:

We have described Kern’s six-step approach and included how it was implemented in this study

Line 108-126

We used Kern’s six-step approach for curriculum development [11]. Step 1 is Problem Identification & General Needs Assessment: our literature review provided the rationale for the curriculum and enabled us to focus on meaningful goals and objectives. The conceptual framework identified and utilized was “situated learning-guided participation” in which didactic and interactive activities facilitate independent learning [12]. Step 2 is the Targeted needs assessment: we identified the specific needs and preferences of our targeted learners which included medical students, residents, faculty and staff and our specific learning environment via group discussions, institutional surveys and informal interviews with several residents and medical students. Step 3 is Goals and Objectives: we developed specific & measurable objectives regarding Impostor Syndrome. We hypothesized that demographic, social, and professional factors may be correlated with Impostor syndrome and that a workshop can improve short-term knowledge and perceptions. Step 4 are the Educational Strategies: to accomplish our educational objectives we identified the appropriate survey tools and developed the interactive curriculum. Step 5 is the Implementation: making the curriculum a reality and converting a good plan into an accomplishment. We identified resources, obtained some institutional support, and developed procedural processes to support the curriculum. Step 6 is Evaluation and feedback, which was accomplished by post intervention knowledge, perception, and behavior-based surveys. The evaluation was done using Kirkpatrick’s framework at Kirkpatrick’s Level 1: Reaction and Kirkpatrick’s Level 2: Learning [13]

Comment:

- The methods section could be split further into approach, data collection etc. Looking at checklists for publication of DELPHI studies might be a way to ensure all the areas are captured.

Response:

 We have reviewed different checklists and adopted sub-titles that seem relevant to the this study.

Subtitles in the Methods section now include:

Study Design:

Curriculum development:

Procedures:

Instruments:

IRB:

Participants:

Analysis:

Comment:

- The Young Imposter Instrument doesn't appear to be clearly referenced at introduction, and it should be made clear whether the tool is validated etc. The explanation is abrupt and appears incomplete- there's no word limit in PLoS One, so try to elaborate concisely to make it clearer what the instrument measures.

Response:

We have expanded the description on the Young Imposter Instrument in Introduction and also expanded further in the discussion section.

Line 67-80:

Valerie Young ED is a leading expert on IS who created the Rethinking Impostor Syndrome™ which has delivered educational solutions including presentations, workshops, and coaching protocols regarding IS to over half a million people around the world since 1982. She developed the Young Impostor Scale (YIS) which is used to dichotomously assess for the presence or absence of IS [5]. YIS is widely used in the lay community and available on the internet. It was recently validated in a study of 138 medical students which revealed that almost a quarter of male medical students and nearly half of female students experienced IS and IS was found to be significantly associated with burnout indices [6]. In her book, “The Secret Thoughts of Successful Women: Why Capable People Suffer from the Impostor Syndrome and How to Thrive in Spite of It”; Dr. Young based on decades of research studying fraudulent feelings among high achievers uncovered five “competence subtypes”—or internal rules that people who struggle with confidence attempt to follow that may be holding them back from achieving their full potential. The competence subtypes are the perfectionist, the super-person, the natural genius, the soloist, and the expert [7].

Line 379-381:

However, it has been noted that there is a lack of normative data for the Young Impostor Scale and that a normal or acceptable level of IS has not been delineated [6].

Comment:

- The way in which the discussion section is written could also be tightened up. I think the framing pushes into causality and takes a positivist approach to the constructs. 

It's not clear how the too brief theoretical framework described in the methods was applied in the interpretation of the results and I guess that's where I can see a bit of dissonance between stated epistemology/methodology and what appears to be a more positivist lens?

Response:

We have expanded on the framework in the methods section and have now attempted to described in the first paragraph, how the methods were applied to the results. We have also attempted to introduce some uncertainty and reduce the apparent positivist approach. 

Line 281-289:

In this cross-sectional study, we developed a curriculum for an Impostor Syndrome educational interactive workshop using the Kern’s six-step approach. We used the framework grounded in situated learning and guided participation to develop interactive and engaging activities including small and large group discussion sessions to promote interests in the concepts and principles regarding IS, support independent learning and encourage problem solving. However, we do not have any longitudinal data to confirm that the independent learning continued over time. We also evaluated our program only at Kirkpatrick’s reaction and learning levels; evaluations at Kirkpatrick’s behavior level (the degree to which participants apply what they learned during training when they are back on the job) may be more useful. 

Comment:

 I can't help but feel like a greater description of what the results mean is necessary before jumping into the assertion of what it demonstrated, and the opening line of the discussion seems displaced. I acknowledge that this is a stylistic issue but we want your manuscript to be the highest quality and have the highest impact it can, and overall I feel the discussion can be improved, made more robust follow a more structured approach and argue for the conclusion you land on, piecing the collected evidence together with theoretical framework and available wider literature in discussion. 

Response:

The noted opening line has been removed. We have provided a greater description of the results. 

Examples of description of results followed by some discussion include:

Line 290-295:

In this medical education cohort, more than half of the participants (57%) were positive for Impostor syndrome. These individuals compared to those negative for IS had higher perceptions of the adverse impact of IS on reaching their full potential and professional relationships. Baseline scores on the IS Knowledge Survey for all participants who participated in the workshop also increased significantly from pre-intervention mean of 4.94 to post-intervention mean of 5.78.

Line 305-313:

The percentage of students meeting criteria for the impostor syndrome in our study was not significantly different between males and females. This is consistent with the literature; whereas the earlier literature on impostor syndrome focused on women, half of the included studies in a recent systemic review found no difference in the rates of men and women suffering from impostor syndrome [3]. In our study there were no significant differences between IS rates in UiM (56%) versus non-UIM participants (44%). Previous studies have demonstrated that IS was more common among underrepresented in medicine racial groups, with impostor feelings significantly negatively correlated with psychological wellbeing and positively correlated with depression and anxiety. [2,3,15].

Line 320-333:

In this study, the prevalence rate of IS of approximately 64% for GME administrators was higher than that for physicians/medical students at 54% but was not statistically significantly different. GME administrators identified significantly higher rates of contributory risk factors for IS including the negative impact of IS on personal and professional relations. This data adds to a recent study demonstrating that 74% of family medicine residency coordinators had moderate to high burnout rates with a significant negative correlation between overall job satisfaction and burnout scores (16). In our study, GME coordinators significantly identified career changes as a significant contributory risk factor to IS. Career changes suggests increased turnover rates which has been shown to be negatively correlated with job satisfaction, and burnout. (16). This data demonstrates that GME administrators are at increased risk of IS similar to faculty, residents, and medical students. Therefore, GME coordinators/administrators should be actively and strategically included in institutional and GME wellness programs in order to increase work satisfaction and decrease burnout rates.

Lines 348-355:

Overall, Impostor competence sub-types significantly correlated with adverse perceptions such as stress, limitation of full potential, negative family, and team relations. Contributory risk factors were also significantly associated with competence subtypes. Parent expectations and female gender were three times more likely to be associated with those with perfectionist competence types, while mental health issues were also three times more likely in those scoring positive for the genius subtype. GME administrators had significantly higher scores for soloist, genius, and perfectionist, whilst physician/medical students had significantly higher scores for expert and super-person competence subtypes.

Comment:

- Lastly, I feel that there is a bit of a tendency to lean towards discreet groups and concepts and that the uncertainty in measurement and in the construct, are not well reflected in the writing.

Response:

We have tried to better reflect the uncertainty in measurement and in the construct, and noted this as a limitation:

Line 286:

However, we do not have any longitudinal data to confirm that the independent learning continued over time.

Line 287:

We also evaluated our program only at Kirkpatrick’s reaction and learning levels; evaluations at Kirkpatrick’s behavior level (the degree to which participants apply what they learned during training when they are back on the job) may be more useful. 

Line 299:

Our findings of a prevalence of 57%, is slightly higher than previously published. This may be due to our small population size, different diagnostic criteria, or specific population characteristics.

Lines 351-359:

However, there were some incongruencies with YIS tool since only three of the five IS competence subtypes (expert, super person, and perfectionist) significantly correlated with scoring positive for IS in this cohort. Therefore, we consider our findings on competence subtypes as preliminary and pilot that would require confirmation from future studies. Furthermore, even though the competence subtypes instrument tool is in extensive use in the lay community nonetheless we could not find any published validation data. Our findings even though introductory and requiring confirmation have practical implications since there are different strategies, counseling, and practical interventions to combat and mitigate each competent subtype [[5, 7].

Lines 383-385:

However, it has been noted that there is a lack of normative data for the Young Impostor Scale and that a normal or acceptable level of IS has not been delineated [6].

Lines 394-396:

It is possible that our small sample size may have resulted in a type 2 error which may explain the non-significant findings between IS and underrepresented in medicine racial groups, gender, soloist, or genius competence subtypes

Lines 399-401:

It is also possible that our discussions and approach may reveal a more positivist lens bias to the constructs and does not adequately describe the uncertainty in measurement and in the construct.

With specific regards to the criteria for publication:

1. The study appears to present the results of original research.

2. Results reported do not appear to have been published elsewhere.

3. Experiments, statistics, and other analyses are performed to a reasonable standard though I feel the description is lacking in completeness. A structured reporting tool would help with this.

Response:

The CROSS checklist has been utilized 

Comment:

4. Conclusions are not really presented in the best fashion, they appear abruptly at the onset of the discussion and so don't develop a logical argument leading into the conclusion; conclusions are presented causally, which I appreciate is both a stylistic and epistemologic issue, but worth considering as you approach your revision.

Response:

The conclusion section has been modified. We feel that in the discussion we have now developed the logical argument that leads to the discussion. We feel that we have now given a more cautious interpretation of results, based on potential biases and imprecisions and suggest areas for future research.

Lines 403-413:

In conclusion, our study demonstrated that Impostor syndrome was prevalent within this medical education cohort, and that a reflective interactive workshop was useful in increasing knowledge and awareness regarding IS. Our study further adds to the literature by identifying significant differences between academic administrators and physicians/medical students; analyzing self-identified contributory factors and providing data on the IS competence subtypes. Future studies on Young Impostor Scale and the IS competence subtypes are required to better clarify the validity and utility of these instruments. This interactive and reflective workshop has the potential to be applied to other medical education settings. This information may enable future targeted IS interventions for specific groups within medical education community and overall lead to development of effective tools to mitigate Impostor syndrome impact and sequalae amongst sufferers especially in the clinical learning environment. 

Comment:

5. The article is presented in an intelligible fashion and is written in standard English. It is at times a little clumsy and this could be improved by ensuring continuity of thought and logic through the writing.

Response:

We have attempted to improve the quality of the writing by ensuring continuity of thought and logic through the manuscript. 

6. An IRB statement is present.

Comment:

7. The article adheres to appropriate reporting guidelines and community standards for data availability, insofaras there aren't specific reporting guidelines that I'm aware of. However, as mentioned above I think a structured reporting tool could help flesh out the methods a bit better in particular. The study captures qualitative ideas but uses survey methods. The checklists I suggest looking at are CROSS (https://www.equator-network.org/reporting-guidelines/a-consensus-based-checklist-for-reporting-of-survey-studies-cross/), SRQR (https://www.equator-network.org/reporting-guidelines/srqr/) and COREQ (https://www.equator-network.org/reporting-guidelines/coreq/). I would specifically suggest leveraging CROSS, and including the completed form as a supplementary item.

Response:

We have completed CROSS checklist 

Comment:

I look forward to your resubmission. It's great to read a study that delves into some of the complexity of medical training and an important psychological element of it. I could see the study being repeated in the same cohort later in their career, or in a post-graduate cohort to try track how these observations change with changes in seniority- interns, residents, fellows and consultants/attending.

Response:

We thank the editor for his favorable and encouraging comments. We hope that the editor finds our revised manuscript acceptable. 

Reviewers' comments:

Reviewer's Responses to Questions

Comments to the Author

1. Is the manuscript technically sound, and do the data support the conclusions?

Reviewer #1: Yes

2. Has the statistical analysis been performed appropriately and rigorously? 

Reviewer #1: No

Response:

We have included standard deviations as requested by the reviewer.

3. Have the authors made all data underlying the findings in their manuscript fully available?

Reviewer #1: Yes

4. Is the manuscript presented in an intelligible fashion and written in standard English?

Reviewer #1: No

5. Review Comments to the Author

Comment:

Reviewer #1: First, the writing needs to be improved, see sentence below and "affected" should be used instead of "impacted." SEM is not an appropriate statistical tool for a single study; SD needs to be used. I note an extreme sex/gender ratio bias in those completing the survey; is this reflective of the employee base of the institutions studied? Looking at this sentence:

Response:

We have removed all SEM and included SD as per the request of the reviewer.

Yes, the extreme sex/gender ratio bias is especially reflective of GME coordinators, and to some extent medical students and residents. 

Comment:

"Furthermore, there were racial differences with administrators more likely to be underrepresented minority groups which may be reflection of systemic and institutional inequities and access to higher education."

Response:

This sentence has been deleted.

Comment:

I realize that different institutions have different employee cohorts, but most medical schools that I am aware of have an administration composed of MDs, PhDs, Eds, etc., these are the highest paid and highest-ranking members of the institution, and make the decisions in running the institution. It is unclear how being a high status, highly paid administrator is reflective of inequities, unless that is meant to describe those employees who are not administrators. The sentence is also poorly written - "reflection" should be "reflective."

Response:

We apologize for this confusion. Administrators in this manuscript refer to residency program coordinators. This has been better clarified in the manuscript as follows: 

Lines 315-318:

GME administrators work behind the scenes in residency programs providing crucial logistical, and administrative support to residents, program directors, faculty, and staff; and maintaining databases and processes to meet Accreditation Council of Graduate Medical Education (ACGME) accreditation.

Comment:

For the reader not well acquainted with these terms, can it be explained why positive terms like "genius" etc. (that would seem to be associated with high self-esteem) are associated with IS? Finally, a limitation of these studies in general is a lack of correlation with subjective self-evaluation and actual performance. IS implies that the person falsely believes that they are not competent. Whether or not this is true or false is not determined.

Response:

Lines 76-80

Dr. Young based on decades of research studying fraudulent feelings among high achievers uncovered five “competence subtypes”—or internal rules that people who struggle with confidence attempt to follow that may be holding them back from achieving their full potential. The competence subtypes are the perfectionist, the super-person, the natural genius, the soloist, and the expert [7].

Lines 141-148:

Participants further completed a survey to identify their competence subtypes (super-person, soloist, natural genius, expert, perfectionist), which is how individuals with IS may perceive competence. Experts identify competency as ability to master tasks quickly. Soloists identify competency as able to master tasks without seeking help or guidance from others. Geniuses question their competency if they make any small mistake while completing tasks. Super-persons question their competency if they are unable to succeed in every single aspect of their life. Perfectionists will only attempt tasks that they are certain they can master and can execute flawlessly.

Lines 334-338:

Competence subtypes described by Dr. Valerie Young which include expert, super person, perfectionist, soloist and genius refer to the unconscious rule of how individuals define what it means to be competent. Dr. Young explained that all 5 competence subtypes hold an extreme view of competence, with no perception of a competence middle ground and if the person is not operating at the top of their game 24/7, they view themself as incompetent 

Again, thank you for the opportunity to be considered for publication in PLOS

We hope the reviewer is satisfied wth our efforts and will provide a favorable response.

 Respectfully submitted

Dotun Ogunyemi

Dotun Ogunyemi, MD

---

## [Editor Report · Decision Letter 1]

21 Jul 2022

Improving wellness: defeating imposter syndrome in medical education using an interactive reflective workshop

PONE-D-22-00137R1

Dear Dr. Ogunyemi,

We’re pleased to inform you that your manuscript has been judged scientifically suitable for publication and will be formally accepted for publication once it meets all outstanding technical requirements.

Kind regards,

Dylan A Mordaunt, MD, MPH, FRACP

Academic Editor

PLOS ONE

Additional Editor Comments (optional):

Thank you for your resubmission. This now meets the criteria for publication.
---

## [Editor Report · Acceptance letter]

26 Jul 2022

PONE-D-22-00137R1 

Improving wellness: defeating Impostor syndrome in medical education using an interactive reflective workshop 

Dear Dr. Ogunyemi:

I'm pleased to inform you that your manuscript has been deemed suitable for publication in PLOS ONE. Congratulations! Your manuscript is now with our production department. 

Kind regards, 

on behalf of

Associate Professor Dylan A Mordaunt 

Academic Editor

PLOS ONE